# Heat Accumulation in Implant Inter-Osteotomy Areas—An Experimental In Vitro Study

**DOI:** 10.3390/biomedicines11010009

**Published:** 2022-12-21

**Authors:** Shanlin Li, Adam Tanner, Georgios Romanos, Rafael Delgado-Ruiz

**Affiliations:** 1School of Dental Medicine, Stony Brook University, Stony Brook, New York, NY 11794, USA; 2Department of Periodontology, School of Dental Medicine, Stony Brook University, Stony Brook, New York, NY 11794, USA; 3Department of Prosthodontics and Digital Technology, School of Dental Medicine, Stony Brook University, Stony Brook, New York, NY 11794, USA

**Keywords:** bone drilling, implant osteotomy, implant site preparation, infrared thermographic analysis

## Abstract

To examine the influence of the distance between adjacent implant osteotomies on heat accumulation in the inter-osteotomy area, two experimental groups with 15 pairs of osteotomies in Type II polyurethane blocks were compared: 7 mm inter-osteotomy separations (Group A, *n* = 15) and 14 mm inter-osteotomy separations (Group B, *n* = 15). An infrared thermographic analysis of thermal changes in the inter-osteotomy area was completed. A one-way analysis of variance (ANOVA) and Fisher post-test were used to determine group differences. Higher temperatures were recorded in Group A at the coronal and middle levels compared to the apical level in both groups. The temperature reached max temperatures at T80s and T100s. In Group A, the threshold for thermal necrosis was exceeded. Meanwhile, Group B did not reach the threshold for thermal necrosis. Preparing adjacent implant osteotomies in dense bone with a 7 mm separation between their centers increases the temperature in the inter-osteotomy area, exceeding the threshold for bone thermal necrosis; meanwhile, increasing the distance between osteotomies reduces the thermal accumulation and the risk for thermal necrosis.

## 1. Introduction

Dental implants are an excellent treatment option for partially or fully edentulous patients, given their high survival (94.6% ± 5.97%) and success rates (89.7% ± 10.2%) after ten years in function [1]. In general, dental implant beds are prepared through a series of drilling steps prior to implant insertion; these procedures are also called implant bed or implant site preparation [2,3,4]. The implant bed preparation traditionally requires one or more rotating drills to complete the osteotomy, which also produces local microfractures and temperature elevation [5]. If the levels of local trauma and temperature cannot be controlled, the risk for bone thermomechanical damage might increase [5].

Exposure time and temperature elevation are determinants of the degree of bone damage. If the bone temperature reaches the limit of 47 °C for one minute or more, irreversible thermal necrosis occurs [4,6]. In addition, alkaline phosphatase degrades at temperatures above 56 °C; necrosis of the surrounding tissues can occur when the bone reaches temperatures over 60 °C; and temperatures over 70 °C can produce fulminant bone necrosis [7].

The temperature elevation produced during bone drilling is related to a combination of various parameters, with bone density and the drilling technique being the most relevant [8]. Bone density influences how heat is produced and how temperature dissipates [9,10]. Higher temperatures are produced in dense (cortical) bone compared to softer (trabecular) bone [10] because higher frictional forces are produced when drilling in dense bone. Meanwhile, drilling in soft bone will result in lower frictional forces and less heat generation. Furthermore, the low dissipation rates of the temperature produced in dense bone can explain why it is at a higher risk for osteonecrosis than is soft bone [11].

Considering the drilling technique for the implant bed preparation, parameters such as thrust force, feed rate, irrigation, drilling speed, and operator experience have been investigated [12,13,14,15,16,17,18]. Thrust force and feed rate are influenced by the operator, and it seems that novices produce higher forces and feed rates compared to experts, resulting in higher temperatures during bone drilling [12,13]. Irrigation is essential for controlling the temperature during high-speed drilling. Specifically, external irrigation is more efficient for cooling than internal irrigation; high volumes of irrigation reduce the temperature more than low volumes of irrigation [14]; and reduced coolant temperature controls the thermal increase better than higher coolant temperatures [15,16]. If the coolant cannot reach the targets (drill surface and bone), the temperature elevation is not controlled. This occurs when irrigation lines are obstructed or in guided surgery, when the surgical guide and metallic sleeves impede the contact of the coolant with the drill and the bone [17].

Finally, when comparing conventional drilling speeds (1200 rpm–2000 rpm) with slow drilling speeds (50 rpm, 150 rpm, or 300 rpm), it was observed that slow drilling speeds resulted in minimal temperature elevation in soft and dense bone [18]. This was confirmed by a recent systematic review that showed that slow-speed drilling produced minimal temperature increases and similar osseointegration and crestal bone loss compared to what was observed in conventional drilling with irrigation [19].

In relation to sequential osteotomies (one osteotomy adjacent to another) there are not reports in the dental field, and there is only one study by Palmisano et al. [20] that evaluated the heat accumulation phenomenon in orthopedic surgery when multiple adjacent osteotomies are prepared. In their study, it was observed that drilling nine adjacent osteotomies in sequence increased the temperature in the inter-osteotomy space after the fifth osteotomy, and the highest temperatures were measured during the ninth osteotomy [20].

In implant dentistry, the thermal effects of bone drilling are commonly evaluated at the site of the implant osteotomy, but a significant lack of knowledge remains concerning the cumulative thermal effect of drilling adjacent implant osteotomies in Type II bone, where the risk of thermal damage is increased. Specifically, what is the impact of the distance between osteotomies on the heat generated in the inter-osteotomy area of sequentially prepared dental implant osteotomies?

This study aimed to test the null hypothesis, stating there is no difference in temperature accumulation between adjacent implant osteotomies separated by 7 or 14 mm, against the alternative hypothesis that there are differences in temperature accumulation between adjacent implant osteotomies separated by 7 or 14 mm.

## 2. Materials and Methods

### 2.1. Sample Size and Calibration

Sample size for this in vitro study was determined as fifteen pairs of osteotomies per group A and B (*n* = 15). The sample size was estimated for a confidence level of 95% and a confidence interval of 25% using the sample size application from StatPlus: Mac, (Analyst Soft Inc., Walnut, CA, USA)- statistical analysis program for macOS. Version v8.

The osteotomies were prepared by two calibrated operators using an implant motor Frios^®^ S/I connected to a contra-angle WS-75 (Friadent, Dentsply Sirona, Bürmoos, Austria). Tapered implant drills Ref. HIKELT-5-3810 (Bioner Sistemas Implantologicos, Barcelona, Spain; Ref. HIKELT-5-3810) with 3.8 mm diameter and 10 mm length were used. Solid close-cell polyurethane blocks (Sawbones, Pacific Research Labs, Vashon, WA, USA) with a density of 0.64 g/cm^3^ and 40 PCF (pounds per cubic foot) were used for this experiment. The thermal conductivity of this type of block (0.47 W/mK) is comparable to Type II dense bone [21,22,23].

### 2.2. Experimental Setup

Sixty simulated osteotomies, divided into fifteen pairs with 7 mm separation in Type II blocks (Test A, *n* = 15), and fifteen pairs of simulated osteotomies with 14 mm separation in Type II blocks (Test B, *n* = 15) were created. Polyurethane blocks were fixed in a vise oriented with their major diameter parallel to the floor and the border of the surface for evaluation facing upwards. The top side of the blocks that was facing the camera served as reference for marking a line parallel to the edge of the block. The line was traced with a pencil at a 4 mm distance from the edge of the blocks. The centers of the future osteotomies were marked on the line, maintaining 7 mm or 14 mm inter-osteotomy distance depending on the experimental group.

### 2.3. Thermal Analysis

One infrared thermographic camera FLIR A325sc (FLIR Systems Inc, Nashua, NH, USA) equipped with macro lenses FLIR T97215 (FLIR Systems Inc, Nashua, NH, USA) was oriented to the inter-osteotomy area of the future osteotomies (aligned with the osteotomy marks completed previously). This allowed for the recording of the inter-osteotomy area temperatures and produced thermal maps at the surface of the blocks. The camera orientation was adjusted to include in the recordings the top of the block and an additional 15 mm above the block (Figure 1).

The recording parameters were room temperature at 21 degrees Celsius, relative humidity of 50%, and focal distance of 7 cm. The blocks were used in dry conditions, and no irrigation was used during the simulated drilling. Each recording of the thermographic camera started before initializing any implant osteotomy preparation. Before the drills contacted the blocks, a calibration recording of the block’s temperature was documented as a baseline. The temperature was recorded in degrees Celsius. Afterwards, using the software FLIR Research Studio Professional Edition, a vertical line of 10 mm was placed in the inter-osteotomy area. Then, three equidistant lines perpendicular to the first vertical line were drawn at the coronal, middle, and apical levels. Thus, each intersection between the vertical and the horizontal lines evaluated the inter-osteotomy area temperature at the coronal, middle, and apical levels (Figure 2).

The infrared thermographic camera was set in continuous video-capture mode to register the temperature changes at the unit sample (each unit sample comprised a pair of adjacent osteotomies evaluated during a period of ±120 s). The changes produced during drilling at the first and second implant sites and temperature accumulation in the inter-osteotomy area were analyzed in standardized measurements at 20 s, 40 s, 60 s, 80 s, and 100 s.

### 2.4. Statistical Analysis

Statistical analysis was completed using the statistical software Minitab web app. The normality of the data was evaluated using the Kolmogorov–Smirnoff test. ANOVA test was completed. Fisher post-test was used to evaluate the temperature differences in the coronal, middle, and apical areas at 7 mm and 14 mm and at 20 s, 40 s, 60 s, 80 s, and 100 s. Significance was set as *p* < 0.05.

## 3. Results

The temperature increased gradually during and after the preparation of the first and second osteotomies and reached peaks between 60 s and 100 s. The temperature accumulation toward the centers of the inter-osteotomy areas was higher at 80 and 100 s (Figure 3).

### 3.1. Coronal Temperature for 7 mm and 14 mm Inter-Osteotomy Separations

Higher temperatures were observed at the coronal level in 7 mm inter-osteotomy separations compared to 14 mm inter-osteotomy separations. Specifically, at 7 mm, the temperature reached peak values at 60 s (58.86 °C ± SD 19.32 °C). Meanwhile, at 14 mm, peak values were reached at 100 s (28.307 °C ± SD 1.52 °C), as shown in Table 1. Figure 4 illustrates mean temperatures and standard deviations recorded in the coronal region at different time points and different inter-osteotomy distances (Table 1 and Figure 4).

### 3.2. Middle Temperature for 7 mm and 14 mm Inter-Osteotomy Separations

Higher temperatures were observed at the middle level for 7 mm compared to 14 mm inter-osteotomy separations. Specifically, at 7 mm, the temperature reached the higher peak values at 100 s (52.60 °C ± SD 17.39 °C) compared to the peak values for 14 mm distances at 100 s (28.252 °C ± SD 1.972 °C), as shown in Table 2. Figure 5 demonstrates the mean temperatures and standard deviations recorded at the middle region of the inter-osteotomy area at each time point and distance (Table 2 and Figure 5).

### 3.3. Apical Temperatures for 7 mm and 14 mm Inter-Osteotomy Separations

Higher temperatures were observed at the apical level of the inter-osteotomy zone for 7 mm compared to 14 mm separations. Specifically, at 7 mm, the temperature reached the higher peak values at 100 s (28.89 °C ± SD 5.34 °C). At the 14 mm inter-osteotomy distance, peak values were lower at 100 s (23.025 °C ± SD 0.79 °C) (Table 3). Figure 5 demonstrates the peak values of the 15 measurements recorded at each time point and distance (Table 3 and Figure 6).

### 3.4. Statistical Comparisons

Fisher tests showed higher temperatures in the inter-osteotomy area for 7 mm compared to 14 mm separation at 40, 60, 80, and 100 s. (Table 4).

Similarly, the Fisher test showed higher temperatures in the middle region of the inter-osteotomy zones for 7 mm compared to 14 mm separations (Table 5).

In the apical region of the inter-osteotomy zone, statistical differences between 7mm and 14mm were observed at 60, 80, and 100 s (Table 6).

## 4. Discussion

This study aimed to test the null hypothesis, stating there is no difference in temperature accumulation between adjacent implant osteotomies separated by 7 or 14 mm, against the alternative hypothesis that there are differences in temperature accumulation between adjacent implant osteotomies separated by 7 or 14 mm. The null hypothesis was rejected. Higher temperature accumulation was observed in shorter separations (7 mm between osteotomies), surpassing the threshold for thermal necrosis in the coronal and middle zones. In implant dentistry, the thermal analysis of bone drilling is centered in the evaluation of the temperature changes at the osteotomy sites, and existing evidence indicates that frictional forces, drilling forces, drilling speed, and bone density all contribute to thermal changes during implant osteotomy [24,25,26].

The direct consequences of the bone overheating are either bone necrosis, increased bone resorption, or implant failure [27,28,29]. In addition, the effects of increased temperature on bone can be influenced by the phenomena of temperature dissipation and temperature accumulation [20,30,31,32,33]. Temperature dissipation in bone can be observed as the heat transfer from one area to another, and temperature accumulation can be observed as the summation of heat dissipated from more than one heat source [21,26]. When multiple adjacent implant osteotomies are required, the risk for thermal necrosis of the inter-osteotomy area should be evaluated because this parameter could explain interproximal bone resorption. The results of the present study showed that 7 mm of separation produced higher temperatures in the inter-osteotomy area than did a 14 mm separation.

This agrees with the studies in orthopedic surgery completed by Gholampour et al., 2019 [34], who prepared adjacent osteotomies at 6 mm, 12 mm, and 16 mm separations in femoral bone. Thermocouples and an infrared camera were used to evaluate the effects of a coolant and separation on the temperature at the first osteotomy site. The results showed that 6 mm separation resulted in higher temperatures, and increasing the time between drilling and the use of coolant limited the temperature increase.

The results of the present study are also in agreement with Palmisano et al., 2015 [20], who investigated the heat-accumulation phenomenon in sequential orthopedic drilling. In their study, nine sequential osteotomies were prepared in a 3 × 3 array on cadaver tibia, using three different drill types. Temperature changes were recorded at the center of four adjacent osteotomies using thermocouples. Their findings demonstrated that the temperatures were higher after the last osteotomy compared to the temperatures after the first osteotomy, and they demonstrated that the heat accumulation and heat dispersion increased when adjacent preparations were completed in dense bone.

In the present study, the results showed that the coronal and middle levels of the inter-osteotomy area presented the highest temperatures at times T4 and T5. Meanwhile, a minimal thermal increment was detected at the apical levels at all the evaluated times.

These findings can be explained by three factors. The first factor is the continued contact of the drills with the cortical compared to the middle and apical. The second factor is the tapered drill design toward the apex, and the third factor is the larger distance between adjacent osteotomies at the apical. This agrees with the study by Heydari et al., [35], who observed higher temperatures during drilling in dense bone due to increased fiction and heat accumulation from the increased time that the drills were in contact with the bone.

In the present study, the distances of 7 mm or 14 mm between the centers of the osteotomies were used, assuming that either adjacent implants with regular diameters (3.8 mm–4 mm) are inserted or adjacent implants are inserted but include a space for a pontic for the fabrication of a bridge. It has been recommended to maintain a distance of 3 mm between adjacent implants because it can preserve the bone and gingival papilla height. When the distance between adjacent implants is reduced to 2 mm or 1 mm, the bone remodeling between implants increases, resulting in bone loss. In addition, when the distance between adjacent implants is less than 3 mm, angiogenesis is reduced. Additionally, the number of blood vessels decreases, and the bone formation is impaired [36,37,38,39]. Within the limitations of this study, it seems also reasonable to think that distances smaller than 3 mm can also produce increased temperature accumulation and increased risk of osteonecrosis in adjacent osteotomies.

Limitations of this study include: only one type of polyurethane block was used, and given the homogeneous characteristic of the blocks, the results can be extrapolated only to bone with similar conditions (dense bone); only one type of drill was used, and the effect of other drill designs remains unknown; only two inter-osteotomy distances were evaluated, and therefore, the information related to the thermal behavior when drilling adjacent osteotomies with shorter or larger separations is missing.

The strengths of this study include: the use of a controlled experimental setup, calibrated operators, and the utilization of bone blocks with a well-known thermal coefficient comparable to human dense bone. This study alerts clinicians to the thermal risks of drilling adjacent dental implant osteotomies. In addition, potential methods for reducing thermal accumulation in the inter-osteotomy zone require further investigation.

## 5. Conclusions

Preparing adjacent implant osteotomies in dense bone with a 7 mm separation between their centers increases the temperature in the inter-osteotomy area, exceeding the threshold for bone thermal necrosis; meanwhile, increasing the distance between osteotomies reduces the thermal accumulation and the risk for thermal necrosis.

## Figures and Tables

**Figure 1 biomedicines-11-00009-f001:**
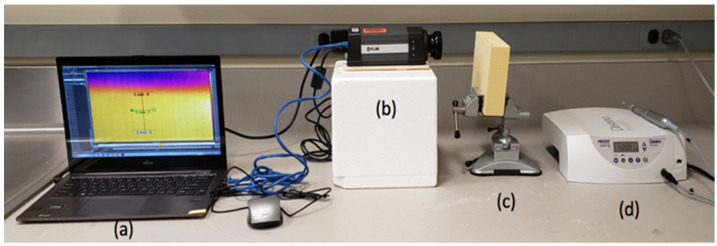
(**a**) Laptop with thermography camera software, (**b**) thermographic camera, (**c**) polyurethane bone block mounted in a vise, and (**d**) implant motor and contra angle.

**Figure 2 biomedicines-11-00009-f002:**
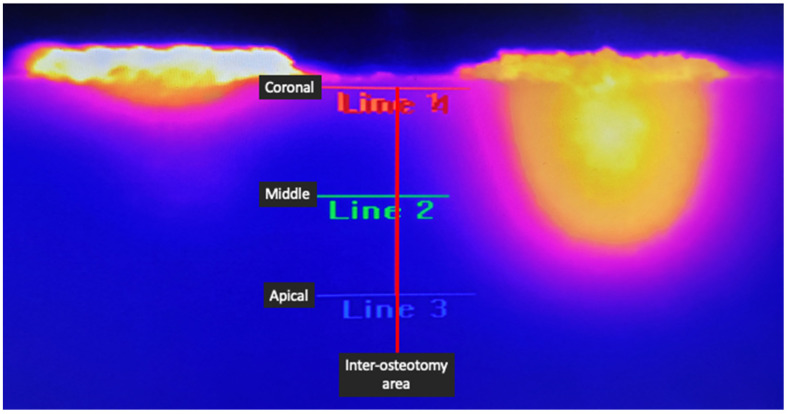
Image demonstrating the setup of measurement recording. The vertical red line in the middle signifies the inter-osteotomy center where the measurements were taken. Lines 1, 2, and 3 indicate the coronal, middle, and apical levels of the inter-osteotomy area, respectively.

**Figure 3 biomedicines-11-00009-f003:**
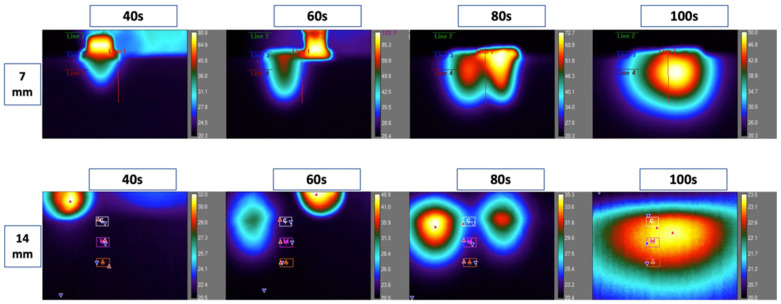
Thermal camera recordings from 7 mm and 14 mm inter-osteotomy distances at 40 s, 60 s, 80 s, and 100 s. This figure illustrates the live recording of the thermal behavior during the preparation of the osteotomies. The red line represents the center between osteotomies. Along the red line, three zones: coronal, middle, and apical. Higher temperatures were recorded at the 7 mm distance compared to 14 mm. Each of the screenshots in the 14 mm recordings shows three letters: C, M, A (coronal, middle, and apical). Each of the screenshots in the 14 mm recordings shows three letters: C, M, A (coronal, middle, and apical). There are also red solid arrows illustrating the hottest region and blue solid arrows illustrating the coldest regions within the zone of evaluation.

**Figure 4 biomedicines-11-00009-f004:**
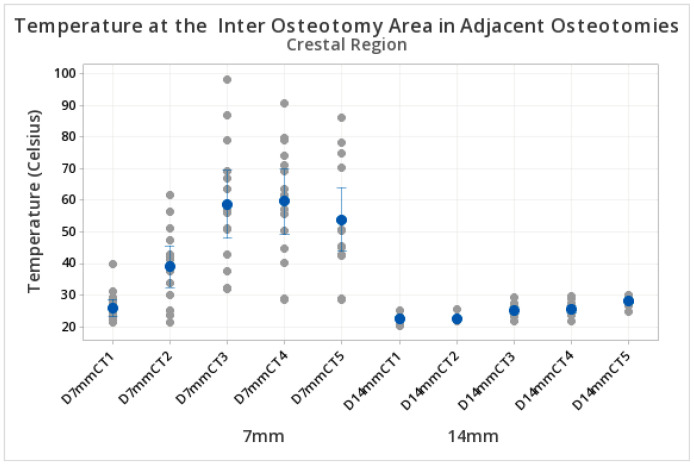
Temperature in the inter-osteotomy area. Measurements at the coronal level. The y axis shows the temperature reached during the preparation of the osteotomies. The x axis indicates the different test groups at 7 mm and 14 mm inter-osteotomy separations at the coronal level. Ascending temperatures were observed in both groups and were influenced by the time and separation between osteotomies. D = distance, C = crestal, T = time (T1 = 20 s, T2 = 40 s, T3 = 60 s, T4 = 80 s, T5 = 100 s). The blue dots represent the central mean value. The grey dots represent the distribution of upper and lower temperatures (sometimes overlapping). The blue lines if present, represent standard deviations.

**Figure 5 biomedicines-11-00009-f005:**
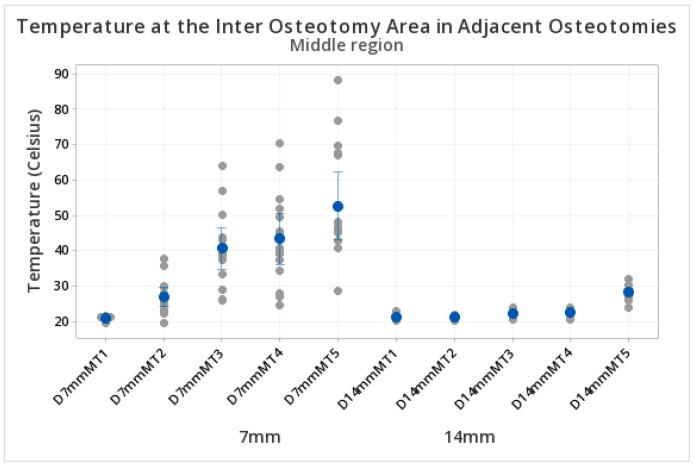
Temperature in the inter-osteotomy area. Measurements at the middle level. The y axis shows the temperature reached during the preparation of the osteotomies. The x axis indicates the different test groups at 7 mm and 14 mm inter-osteotomy separations at the middle level. Ascending temperatures were observed in both groups and influenced by the time and separation between osteotomies. D = distance, M = middle, T = time (T1 = 20 s, T2 = 40 s, T3 = 60 s, T4 = 80 s, T5 = 100 s). The blue dots represent the central mean value. The grey dots represent the distribution of upper and lower temperatures (sometimes overlapping). The blue lines if present, represent standard deviations.

**Figure 6 biomedicines-11-00009-f006:**
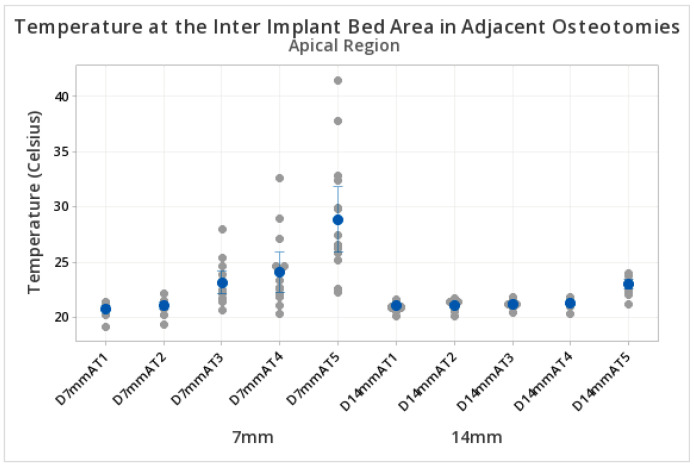
Temperature in the inter-osteotomy area in adjacent osteotomies. Measurements at the apical level. The y axis shows the temperature reached during the preparation of the osteotomies. The x axis indicates the different test groups at 7 mm and 14 mm inter-osteotomy area separations at the apical level. Ascending temperatures were observed in both groups, influenced by the time. D = distance, A = apical, T = time (T1 = 20 s, T2 = 40 s, T3 = 60 s, T4 = 80 s, T5 = 100 s). The blue dots represent the central mean value. The grey dots represent the distribution of upper and lower temperatures (sometimes overlapping). The blue lines if present, represent standard deviations.

**Table 1 biomedicines-11-00009-t001:** Descriptive statistics of inter-implant bed temperature at distances of 7 mm and 14 mm in the coronal area of the dense bone. D = distance, C = coronal, T = time (T1 = 20 s, T2 = 40 s, T3 = 60 s, T4 = 80 s, T5 = 100 s).

Inter-Osteotomy Distance and Time. Coronal Area	Sample Size	Mean	Standard Deviation	95% CI
D7CT0	15	21.133	0.713	(15.976, 26.290)
D7CT20	15	26.03	4.97	(20.87, 31.19)
D7CT40	15	39.08	11.90	(33.92, 44.23)
D7CT60	15	58.86	19.32	(53.70, 64.01)
D7CT80	15	59.74	18.51	(54.59, 64.90)
D7CT100	15	54.07	18.19	(48.91, 59.22)
D14CT0	15	22.040	1.679	(16.883, 27.197)
D14CT20	15	22.597	1.084	(17.440, 27.754)
D14CT40	15	22.655	0.835	(17.498, 27.812)
D14CT60	15	25.195	1.837	(20.038, 30.352)
D14CT80	15	25.692	2.034	(20.535, 30.849)
D14CT100	15	28.307	1.520	(23.150, 33.464)

**Table 2 biomedicines-11-00009-t002:** Descriptive statistics of inter-implant bed temperature at distances of 7 mm and 14 mm in the middle area of the dense bone. D = distance, M = middle, T = time (T1 = 20 s, T2 = 40 s, T3 = 60 s, T4 = 80 s, T5 = 100 s).

Inter-Osteotomy Distance and Time. Middle Area	Sample Size	Mean	Standard Deviation	95% CI
D7MT0	15	20.816	0.484	(17.158, 24.475)
D7M20	15	20.980	0.496	(17.322, 24.638)
D7MT40	15	26.90	4.91	(23.24, 30.55)
D7MT60	15	40.74	10.61	(37.08, 44.40)
D7MT80	15	43.42	13.13	(39.77, 47.08)
D7MT100	15	52.60	17.39	(48.94, 56.26)
D14MT0	15	21.223	0.450	(17.565, 24.881)
D14MT20	15	21.344	0.609	(17.686, 25.002)
D14MT40	15	21.376	0.452	(17.718, 25.034)
D14MT60	15	22.284	0.732	(18.626, 25.943)
D14MT80	15	22.524	0.707	(18.866, 26.182)
D14MT100	15	28.252	1.972	(24.594, 31.910)

**Table 3 biomedicines-11-00009-t003:** Descriptive statistics of inter-implant bed temperature at distances of 7 mm and 14 mm in the apical area of the dense bone. D = distance, A = apical, T = time (T1 = 20 s, T2 = 40 s, T3 = 60 s, T4 = 80 s, T5 = 100 s).

Inter-Osteotomy Distance and Time. Apical Area	Sample Size	Mean	Standard Deviation	95% CI
D7AT0	15	20.740	0.523	(19.757, 21.723)
D7AT20	15	20.843	0.529	(19.860, 21.826)
D7AT40	15	21.132	0.658	(20.148, 22.115)
D7AT60	15	23.194	1.820	(22.211, 24.177)
D7AT80	15	24.131	3.247	(23.148, 25.114)
D7AT100	15	28.89	5.34	(27.91, 29.88)
D14AT0	15	21.092	0.387	(20.109, 22.075)
D14AT20	15	21.112	0.387	(20.128, 22.095)
D14AT40	15	21.148	0.392	(20.165, 22.131)
D14AT60	15	21.2436	0.3480	(20.2604, 22.2268)
D14AT80	15	21.2933	0.3576	(20.3101, 22.2765)
D14AT100	15	23.025	0.790	(22.041, 24.008)

**Table 4 biomedicines-11-00009-t004:** Fisher test for 14 mm vs. 7 mm in the coronal area. * *p* < 0.05.

Coronal Groups Comparison	Difference of Means	SE of Difference	95% CI	T-Value	Adjusted *p* Value
D14CT0–D7CT0	0.91	3.69	(−6.39, 8.20)	0.25	0.806
D14CT20–D7CT20	−3.43	3.69	(−10.72, 3.86)	−0.93	0.354
D14CT40–D7CT40	−16.42	3.69	(−23.72, −9.13)	−4.45	0.000 *
D14CT60–D7CT60	−33.66	3.69	(−40.95, −26.37)	−9.11	0.000 *
D14CT80–D7CT80	−34.05	3.69	(−41.34, −26.76)	−9.22	0.000 *
D14CT100–D7CT100	−25.76	3.69	(−33.05, −18.47)	−6.97	0.000 *

**Table 5 biomedicines-11-00009-t005:** Fisher test for 14 mm vs. 7 mm in the middle area. * *p* < 0.05.

Middle Groups Comparison	Difference of Means	SE of Difference	95% CI	T-Value	Adjusted *p* Value
D14MT0–D7MT0	0.41	2.62	(−4.77, 5.58)	0.16	0.877
D14MT20–D7M20	0.36	2.62	(−4.81, 5.54)	0.14	0.890
D14MT40–D7MT40	−5.52	2.62	(−10.69, −0.35)	−2.11	0.037 *
D14MT40–D7MT60	−19.36	2.62	(−24.54, −14.19)	−7.39	0.000 *
D14MT80–D7MT80	−20.90	2.62	(−26.07, −15.73)	−7.98	0.000 *
D14MT100–D7MT100	−24.35	2.62	(−29.52, −19.17)	−9.29	0.000 *

**Table 6 biomedicines-11-00009-t006:** Fisher test for 14 mm vs. 7 mm in the apical area. * *p* < 0.05.

Apical Groups Comparison	Difference of Means	SE of Difference	95% CI	T-Value	Adjusted *p* Value
D14AT0–D7AT0	0.352	0.704	(−1.038, 1.743)	0.50	0.618
D14AT20–D7AT20	0.268	0.704	(−1.122, 1.659)	0.38	0.704
D14AT40–D7AT40	0.017	0.704	(−1.374, 1.407)	0.02	0.981
D14AT60–D7AT60	−1.950	0.704	(−3.341, −0.560)	−2.77	0.006 *
D14AT80–D7AT80	−2.838	0.704	(−4.228, −1.447)	−4.03	0.00 *
D14AT100–D7AT100	−5.869	0.704	(−7.260, −4.479)	−8.33	0.00 *

## Data Availability

The data from this experiment will be provided after request to the authors.

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
