# Peer review of "Heat Accumulation in Implant Inter-Osteotomy Areas—An Experimental In Vitro Study"

_biomedicines, 2022, doi:10.3390/biomedicines11010009_

Round 1

Reviewer 1 Report

Hypothesis of the study is missing.

What is the rationale of choosing 7 mm and 14 mm?

Conclusion part should be extended.

It is not clear how the results would affect the clinical practice.

A lot of typing errors including missing spaces and superscripts.

Author Response

Answers to Reviewer 1

Dear Reviewer 1, sincere thanks for your expertise and time for reviewing our manuscript. You certainly pointed out some aspects that require clarification. You will find next the point-by-point answers to your comments.

-Hypothesis of the study is missing

In the introduction a paragraph with the hypothesis of the study was included as follows:

“This study aimed to test the null hypothesis of no differences in temperature accumulation between adjacent implant osteotomies separated by 7 or 14mm against the alternative hypothesis of differences in temperature accumulation between adjacent implant osteotomies separated by 7 or 14mm”

-What is the rationale of choosing 7 mm and 14 mm?

Thank you for your comment.

The distances of 7mm or 14mm between the centers of the osteotomies were selected assuming that adjacent implants with regular diameter (3.8mm-4mm) for single crowns are inserted or adjacent implants are inserted including a pontic space for the fabrication of a bridge.

In the present study, the 7mm distance accounts for 3.8mm-4mm implant diameter with 3mm distance between the walls of the osteotomies.  Please see illustrations for clarity. 

The 14mm distance accounts for an additional pontic space between osteotomies (for example when an implant bridge is going to be fabricated). Please see illustration for clarity

In addition, 3mm of separation between implants possess a biological justification because maintains bone and gingival height. A paragraph was included within the discussion section, as well as the references supporting the statements as follows:

“It has been recommended to maintain a distance of 3mm between adjacent implants because can preserve the bone and gingival papilla height. When the distance between adjacent implants is reduced to 2mm or 1mm, the bone remodeling between implants increase resulting in bone loss. In addition, when distance between adjacent implants is less than 3mm, the angiogenesis is reduced, the number of blood vessels decreased, and the bone formation is impaired.”  

Tarnow D, Cho S, Wallace S. The effect of inter-implant distance on the height of inter-implant bone crest. J Periodontol. 2000;71:546-9. 

Ramanauskaite A, Sader R. Esthetic complications in implant dentistry. Periodontol 2000. 2022;88:73-85. 

Gastaldo J, Cury P, Sendyk W. Effect of the vertical and horizontal distances between adjacent implants and between a tooth and an implant on the incidence of interproximal papilla. J Periodontol. 2004;75:1242-6. 

Traini T, Novaes A, Piattelli A, Papalexiou V, Muglia V. The relationship between interimplant distances and vascularization of the interimplant bone. Clin Oral Implants Res. 2010;21:822-9.

-Conclusion part should be extended

Thank you for your comment.

We kept the conclusions as specific as possible to avoid overstating our results.

However, following your recommendations, the conclusions were extended as follows.

“Preparing simulated adjacent implant osteotomies in dense bone with 7mm separation between their centers, increases the temperature at the inter-osteotomy area exceeding the threshold for bone thermal necrosis meanwhile, increasing the distance between osteotomies reduces the thermal accumulation.” 

-What is the clinical relevance of this findings

Thank you for your comment.

The following sentence was added at the end of the discussion section.

“Drilling two adjacent implant beds with separations of 7mm between the center of the osteotomies in dense bone, increase the risk for thermal osteonecrosis at the inter osteotomy zone thanks to the temperature accumulation phenomena.   This study alerts clinicians completing implant surgerie

Reviewer 2 Report

The paper "Heat accumulation in implant inter-osteotomy areas: an experimental in vitro-study" is of some interest. The authors have examinated the influence of distance between adjacent implant osteotomies on heat accumulation at the inter-osteotomy area using polyurethane blocks.

The abstract is clear and concise. The  introduction is quite clear and comprehensive. Materials and methods are well described; many figures and tables help the reader to understand. The discussion is quite clear and well written; the limitations and the strenghts are highlighted in the text.

I think that the paper may be of some help and interest, despite the limitations of the in vitro study, to clinicians.

Author Response

Answers to Reviewer 2

Dear reviewer 2 many thanks for your comments. The authors certainly believe that this in-vitro study possess clinical implications and therefore, a paragraph was added to the discussion section as follows:

"This study alerts clinicians of the thermal risks of drilling adjacent dental implant osteotomies. In addition, potential methods for reducing thermal accumulation at the inter-osteotomy zone require further investigation."

Reviewer 3 Report

The article titled "Heat Accumulation in Implant Inter-Osteotomy Areas. An Experimental In-Vitro Study." is very well written. It's clear and comprehensive. The introduction puts the subject into perspective, and the discussions are pertinent.

Only few minor suggestions from my part:

line 22: The word Conclusions should be removed, as the abstract has no sections.

line 162. "Figure 3  demonstrates the peak values of the 15 measurements recorded at each time point and distance (Table 1 and Figure 4)." Sorry, I don't fully understand the meaning. Please consider reformulating. 

Same for "Figure 4 demonstrates the peak values of the 15 measurements recorded at each time point and distance (Table 2 and Figure 5)." It seems the formulation it's repeating (line 194). Please consider modifying it.

Author Response

Dear Reviewer 3

We appreciate your time, expertise, and comments for our manuscript entitled “Heat Accumulation in Implant Inter-Osteotomy Areas. An Experimental In-Vitro Study”

Here are the point-by-point answers to your comments:

-The article titled "Heat Accumulation in Implant Inter-Osteotomy Areas. An Experimental In-Vitro Study." is very well written. It's clear and comprehensive. The introduction puts the subject into perspective, and the discussions are pertinent.
Many thanks for your comment. 

Only few minor suggestions from my part:
-line 22: The word Conclusions should be removed, as the abstract has no sections.
Thanks for your comment. The word “Conclusions” was removed from the abstract. 

line 162. "Figure 3 demonstrates the peak values of the 15 measurements recorded at each time point and distance (Table 1 and Figure 4)." Sorry, I don't fully understand the meaning. Please consider reformulating. 
Thanks for pointing out this error. The lines 163 to 165 were edited  as follows: 
“Figure 4 illustrates mean temperatures and standard deviations recorded at the coronal region at different time points and different inter osteotomy distances. (Table 1 and Figure 4).”

Same for "Figure 4 demonstrates the peak values of the 15 measurements recorded at each time point and distance (Table 2 and Figure 5)." It seems the formulation it's repeating (line 194). Please consider modifying it.
Many thanks for finding this error. The lines 179 to 181 were edited as follows:
“Figure 5 demonstrates the mean temperatures and standard deviations recorded at the middle region of the inter osteotomy area at each time point and distance (Table 2 and Figure 5).”
